# The Association of Oxaliplatin-Containing Adjuvant Chemotherapy Duration with Overall and Cancer-Specific Mortality in Individuals with Stage III Colon Cancer: A Population-Based Retrospective Cohort Study

Colin Sue-Chue-Lam [1,2], Christine Brezden-Masley [2,3], Rinku Sutradhar [2,4], Amy Y. X. Yu [5] and Nancy N. Baxter [2,4,6,7,*]

1 Department of Surgery, University of Toronto, Toronto, ON M5T 1P5, Canada; colin.sue.chue.lam@alum.utoronto.ca
2 Institute of Health Policy, Management and Evaluation, University of Toronto, Toronto, ON M5T 3M6, Canada
3 Division of Medical Oncology, Sinai Health System, Mount Sinai Hospital, Toronto, ON M5G 1X5, Canada
4 ICES, Toronto, ON M4N 3M5, Canada
5 Department of Medicine (Neurology), University of Toronto, Sunnybrook Health Sciences Centre, Toronto, ON M4N 3M5, Canada
6 Li Ka Shing Knowledge Institute, St. Michael's Hospital, Toronto, ON M5B 1T8, Canada
7 Melbourne School of Global and Population Health, 207 Bouverie St. Level 5, University of Melbourne, Melbourne, VIC 3010, Australia
* Correspondence: baxter@unimelb.edu.au

**Abstract:** Purpose: Few studies have examined the relationship between duration of oxaliplatin-containing adjuvant chemotherapy for stage III colon cancer and mortality in routine practice. We examined the association between treatment with 50% versus >85% of a maximal course of adjuvant therapy (eight cycles of CAPOX, twelve cycles of FOLFOX) and mortality in stage III colon cancer. Methods: Using linked databases, we identified Ontarians aged ≥18 years at diagnosis of stage III colon cancer between 2007 and 2019. In the primary comparison, we compared patients who received 50% or >85% of a maximal course of adjuvant therapy; in a secondary comparison, we evaluated a dose effect across patients who received FOLFOX in one-cycle increments from six to ten cycles against >85% (more than ten cycles) of a maximal course of FOLFOX. The main outcomes were overall and cancer-specific mortality. Follow-up began 270 days after adjuvant treatment initiation and terminated at the first of the outcome of interest, loss of eligibility for Ontario's Health Insurance Program, or study end. Overlap propensity score weights accounted for baseline between-group differences. We determined the hazard ratio, estimating the association between mortality and treatment. Non-inferiority was concluded in the primary comparison for either outcome if the upper limit of the two-sided 95% CI was ≤1.11, which is the margin used in the International Duration Evaluation of Adjuvant Chemotherapy Collaboration. Results: We included 3546 patients in the analysis of overall mortality; 486 (13.7%) received 50% and 3060 (86.3%) received >85% of a maximal course of therapy. Median follow-up was 5.4 years, and total follow-up was 20,510 person-years. There were 833 deaths. Treatment with 50% of a maximal course of adjuvant therapy was associated with a hazard ratio of 1.13 (95% CI 0.88 to 1.47) for overall mortality and a subdistribution hazard ratio of 1.31 (95% CI 0.91 to 1.87) for cancer-specific mortality versus >85% of a maximal course of therapy. In the secondary comparison, there was a trend toward higher overall mortality in patients treated with shorter durations of therapy, though confidence intervals overlapped considerably. Conclusion: We could not conclude that treatment with 50% of a maximal course is non-inferior to >85% of a maximal course of adjuvant therapy for mortality in stage III colon cancer. Clinicians and patients engaging in decision-making around treatment duration in this context should carefully consider the trade-off between treatment effectiveness and adverse effects of treatment.

**Keywords:** colon cancer; adjuvant therapy; oxaliplatin; non-inferiority

## 1. Introduction

For patients with colon cancer undergoing adjuvant therapy, maximizing treatment effectiveness while preventing the dose-dependent peripheral neuropathy commonly caused by oxaliplatin is a top priority [1–3]. To address this, the International Duration of Adjuvant Therapy (IDEA) collaboration conducted an international, multi-center trial to determine the non-inferiority of 3 months to 6 months of adjuvant therapy for survival in patients with stage III colon cancer [4]. Peripheral neuropathy was reduced in the 3-month arm, but the findings were not conclusive of non-inferiority for overall survival and disease-free survival at 5 years for the overall cohort.

The complex results of the IDEA collaboration have been described as 'a Rorschach test for investigators and clinicians', indicating that practical guidance stemming from these results has not been straightforward [5]. For high-risk cancers, some suggest 3–6 months of CAPOX as appropriate, while others recommend 6 months of therapy regardless of regimen [1,6]. Still others provide strong recommendations for 3 or 6 months of CAPOX and 6 months of FOLFOX within the general population of patients with stage III cancers, while cautioning against adapting treatment to risk subgroups [2]. Adding to the difficulty of translating this research into practice is the longstanding issue of patients in trials being younger and less comorbid than those in routine practice [7,8], as well as evolving research showing that chemotherapy dosing differs between trials and routine practice in ways that may impact treatment effectiveness [9].

In the context of this continued uncertainty about the role of IDEA in routine practice, we sought to examine the association of adjuvant therapy duration with mortality in stage III colon cancer using linked population-based and health administrative datasets. To this end, we performed a retrospective cohort study using these linked databases to investigate the association between treatment with a shortened versus a long course of oxaliplatin-containing adjuvant chemotherapy and mortality.

## 2. Methods

### 2.1. Study Design, Inclusion Criteria, and Exclusion Criteria

This retrospective cohort study using routinely collected data is reported in accordance with the RECORD statement (Tables A1 and A2) [10]. Health services and population databases held at ICES (formerly the Institute for Clinical Evaluative Sciences) were used to ascertain covariates, exposures, and outcomes. These datasets were linked using unique encoded identifiers and analyzed at ICES. ICES is an independent non-profit institution with protected legal status, allowing it to analyze routinely collected health service and population data.

We used the algorithms described in Table 1 to identify Ontario residents aged ≥18 years at the time of diagnosis of incident stage III adenocarcinoma of the colon in the Ontario Cancer Registry (OCR) between 1 January 2007 and 31 December 2019. Individuals were excluded for any cancer diagnosis within 5 years prior to colon cancer diagnosis, a prior colorectal cancer diagnosis at any time, multiple simultaneous primary colon cancer diagnoses, no curative resection within 6 months of diagnosis, failing to initiate adjuvant therapy within 16 weeks of surgery, less than 2 years OHIP coverage prior to the date of first adjuvant treatment, and an indeterminate oxaliplatin regimen.

**Table 1.** Results from sensitivity analyses comparing one-cycle increments of FOLFOX against >85% (eleven to twelve cycles) of a maximal course of oxaliplatin-containing adjuvant therapy. Estimates are derived from overlap-weighted Cox and subdistribution hazards models.

| Comparison | HR (95% CI) for Overall Mortality | sHR (95% CI) for Cancer-Specific Mortality |
|---|---|---|
| 10 cycles vs. 11–12 cycles | 1.07 (0.87 to 1.32) | 1.33 (1.05 to 1.70) |
| 9 cycles vs. 11–12 cycles | 1.11 (0.89 to 1.40) | 1.13 (0.86 to 1.49) |
| 8 cycles vs. 11–12 cycles | 1.17 (0.94 to 1.47) | 1.36 (1.04 to 1.76) |
| 7 cycles vs. 11–12 cycles | 1.53 (1.19 to 1.95) | 1.82 (1.37 to 2.42) |
| 6 cycles vs. 11–12 cycles * | 1.14 (0.88 to 1.48) | 1.37 (0.95 to 1.96) |

* Durations used in primary comparison

*2.2. Exposure and Index Date for Follow-Up*

For the primary comparison, we included only patients who received 50% (four cycles of CAPOX or six cycles of FOLFOX) or >85% (seven or eight cycles of CAPOX or eleven or twelve cycles of FOLFOX) of a maximal 6-month course of oxaliplatin-containing adjuvant therapy. These durations were chosen to match the IDEA per protocol analysis populations and approximate the treatment durations actually received in the pooled IDEA population [4,11]. In their pooled population, the median durations of therapy received were 12 weeks (IQR 12–12 weeks or six–six cycles) for 3 months of FOLFOX, 24 weeks (IQR 20–24 weeks or ten to twelve cycles) for 6 months of FOLFOX, 12 weeks (IQR 12–12 weeks or six–six cycles) for 3 months of CAPOX, and 24 weeks (IQR 18–24 weeks or ten to twelve cycles) for 6 months of CAPOX. We used the proportion of therapy rather than duration in months to account for minor deviations from the timeline of the typical dosing schedule (i.e., every 2 weeks for FOLFOX and every 3 weeks for CAPOX).

In a secondary comparison, we evaluated a dose effect by comparing patients who received six, seven, eight, nine, or ten cycles of FOLFOX against those who received ≥85% of a maximal course of FOLFOX (i.e., eleven to twelve cycles, the same referent as the primary comparison). We chose to evaluate only patients treated with FOLFOX for the secondary comparison to simplify dose increments, given the differing treatment schedules for FOLFOX and CAPOX.

Oxaliplatin and fluoropyrimidine therapy data were obtained from sources listed in Table 2. We ascertained adjuvant therapy occurring from the first day of adjuvant therapy administration through 270 days later to assign patients to treatment groups. Based on provincial treatment standards and medical oncologist recommendations, this duration was considered appropriate to capture a patient's complete course of therapy [12–14]. To avoid immortal time bias, time zero of follow-up (the index date) for all patients was set at 270 days after adjuvant initiation [15].

**Table 2.** Numbers of FOLFOX-treated patients included in secondary comparison for dose effect.

| Treatment Cycles | n |
|---|---|
| 11–12 | 2857 |
| 10 | 465 |
| 9 | 345 |
| 8 | 355 |
| 7 | 196 |
| 6 | 329 |

We excluded patients who stopped oxaliplatin for over 16 weeks and subsequently resumed within the exposure window as they were likely being treated for recurrence. Patients who died, lost OHIP coverage, received zero cycles of oxaliplatin, or received more than twelve cycles of oxaliplatin during the exposure window (within 270 days after adjuvant initiation) were also excluded.

*2.3. Outcome*

The primary outcomes were (1) overall mortality and (2) cancer-specific mortality. Vital status was obtained from the Registered Persons' Database. Cause of death was ascertained from the Ontario Registrar General—Death (ORGD). Consistent with prior work, any cancer death was considered cancer-specific mortality [16–18].

*2.4. Covariates*

Patient characteristics included age at diagnosis as a continuous variable, sex, frailty, and The Johns Hopkins ACG® System Version 10 Aggregated Diagnosis Groups (ADG) comorbidity score as a continuous variable [19,20]. Frail patients were identified using the ACG® System Frailty flag [21]. Lookback for frailty and the ADG comorbidity score began at the time of the first adjuvant treatment record and extended back 2 years. The Ontario

Marginalization Index (ONMARG) quintile for area-level deprivation and rural residence were defined at the time of the first adjuvant oxaliplatin treatment record [22].

Cancer characteristics including the date of diagnosis, tumor location (proximal or distal colon), and American Joint Committee on Cancer (AJCC) 8th edition T/N stage were ascertained from the OCR, which employs trained stage analysts to abstract TNM stage data from hospital charts and provincial administrative datasets.

Treatment characteristics were observed prior to the index date and included the interval between diagnosis and surgery in days, postoperative complications within 30 days of surgery, the interval between surgery and initiation of adjuvant therapy in days, the adjuvant regimen (CAPOX or FOLFOX), an oxaliplatin dose reduction > 20% of first dose, and chemotherapy complications resulting in presentation to hospital (Table 2).

### 2.5. Missing Data

The missing indicator method was used for the missing deprivation quintile, which was assumed to be missing not at random [23]. We analyzed complete cases when rurality and AJCC T/N stage were missing. In a sensitivity analysis, we assumed they were missing at random and used multiple imputation via chained equations to handle these missing covariates.

### 2.6. Statistical Analysis

Distributions of baseline characteristics were summarized using means for continuous variables and proportions for categorical variables. Standardized differences (SD) quantified imbalance between treatment groups.

For the primary comparison between 50% and >85% of a maximal course of adjuvant therapy, we accounted for confounding by baseline differences between groups using overlap weights. Overlap weights were derived from the propensity score (PS) for treatment, obtained from logistic regression on treatment group and including all baseline characteristics [24–26]. We used an overlap-weighted Cox proportional hazards model to determine the hazard ratio (HR) and two-sided 95% confidence interval (CI) for the association between treatment with 50% versus >85% of a maximal course of adjuvant therapy and overall mortality [27]. We used overlap-weighted Fine and Gray regression models to determine the subdistribution hazard ratio (sHR) and two-sided 95% CI for cancer-specific mortality, treating non-cancer mortality as a competing risk [28]. For the secondary comparison exploring a dose effect, we used the same overlap weighting approach to conduct separate comparisons of patients who received 6, 7, 8, 9, or 10 cycles of FOLFOX against those who received >85% of a maximal course of FOLFOX.

Follow-up began at the index date and terminated at death or censoring. Patients were censored when they lost OHIP eligibility or reached the end of follow-up, whichever came first. The end of follow-up for the analysis of overall mortality was 28 February 2022. Because cause of death data were not available beyond 30 November 2018, this was the end of follow-up for the analysis of cancer-specific mortality. Overlap-weighted cumulative incidence functions (CIF) were estimated to illustrate the risk for each outcome over time. Robust variance estimators accounted for overlap weighting [29].

We conducted non-inferiority analyses for overall mortality and cancer-specific mortality in the primary comparison of 50% versus >85% of a maximal course of adjuvant therapy in the overall cohort. Non-inferiority was declared if the upper limit of the two-sided 95% CI was ≤1.11 (the IDEA overall survival non-inferiority margin). The upper limit of the CIs for both overall mortality and cancer-specific mortality exceeded the non-inferiority margin. We subsequently used two-sided 95% CI to assess the superiority of >85% to 50% of a maximal course of adjuvant therapy.

In exploratory analyses, we determined the associations between treatment with 50% versus 85% of a maximal course of therapy, and each outcome in the following subgroups, using the IDEA subgroups where data were available: age (≤70 years or >70 years), sex (male or female), cancer risk group (T1–T3 and N1 versus T4, N2, or both), tumor location

(proximal or distal colon), and adjuvant regimen (FOLFOX or CAPOX). The PS and overlap weights were re-estimated within the strata of each subgroup. Because these analyses were exploratory, we did not perform formal statistical testing [30].

The threshold for significance was set at one-tailed $p < 0.025$ for non-inferiority comparisons and two-tailed $p < 0.05$ for the superiority comparisons. All analyses were conducted using SAS Enterprise Guide, version 7.1 (SAS Institute Inc., Cary, NC, USA) and R (R Foundation for Statistical Computing, Vienna, Austria).

### 2.7. Sensitivity Analyses

To examine our assumptions about missing data mechanisms, we used multiple imputation by chained equations for missing rurality, AJCC T stage, and AJCC N stage [31]. We included in the imputation model all variables in the PS, an indicator variable for the outcome, and survival time transformed using the cumulative hazard function [32]. Five datasets were imputed [33], overlap weights were calculated in each imputed dataset, and overlap-weighted estimates for survival from within each imputed dataset were then pooled using Rubin's rules [34]. We also examined the effect of treatment duration among the subset of patients who had no dose reduction and were likely more similar in their ability to tolerate chemotherapy. Lastly, we analyzed overall mortality excluding patients diagnosed after 2017 to limit our comparison to the pre-IDEA period.

### 3. Results

After exclusions, we analyzed 3546 patients for the complete case analysis comparing 50% versus >85% of a maximal course of oxaliplatin-containing adjuvant therapy for overall mortality (Figure 1 and Table 3). For the primary comparison, four hundred and eighty patients (14.7%) received 50% of a maximal course and 3060 (86.3%) received >85% of a maximal course of adjuvant therapy. Numbers of patients included in the secondary comparison for a dose effect are included in Table 2. In the unweighted cohort for overall mortality, those who received 50% of a maximal course were older (mean 62.0 years versus 60.1 years, SD 0.19), less often had high-risk cancers (25.3% versus 54.4%, SD 0.62), and were less likely to receive FOLFOX (67.7% versus 93.4%, SD 0.69) than those who received >85%.

Median follow-up for overall mortality was 3.2 years for patients treated with short-duration chemotherapy and 5.8 years for patients treated with long-duration chemotherapy, with a total of 20,510 person-years of follow-up. There were 833 deaths over this follow-up period. For cancer-specific mortality, median follow-up was 2.3 years for patients treated with short-duration chemotherapy and 3.8 years for patients treated with long-duration chemotherapy, with a total of 11,853 person-years of follow-up. There were 442 cancer deaths and 79 non-cancer deaths over this follow-up period.

In the overlap-weighted cohort for both overall mortality and cancer-specific mortality, the distribution of weights was appropriate, all standardized differences were <0.1, and plotted covariate distributions in each treatment group were qualitatively similar, indicating the PS was adequately specified (Figures A1–A4).

Cumulative incidence functions for overall mortality and cancer-specific mortality are plotted in Figures 2 and 3, respectively. The overall mortality risk over 5 years was 16.2% (95% CI 12.6–20.3) among patients treated with 50% of a maximal course of therapy versus 14.8% (95% CI 13.0–16.7) among patients treated with >85% of a maximal course. The HR for the association of treatment with 50% versus >85% cycles of oxaliplatin-containing adjuvant chemotherapy and overall mortality was 1.13 (95% CI 0.88–1.45, non-inferiority $p = 0.55$, superiority $p = 0.34$).

**Table 3.** Distributions of baseline clinical and demographic characteristics of the cohort for overall mortality stratified by treatment. In the weighted cohort, patients are weighted by the overlap propensity score weight.

| Characteristic | Unweighted Cohort, No. (%) | | | | Weighted Cohort, % | | |
|---|---|---|---|---|---|---|---|
| | All (N = 3546) | 50% of Maximum Cycles (N = 486) | >85% of Maximum Cycles (N = 3060) | Std Diff | 50% of Maximum Cycles | >85% of Maximum Cycles | Std Diff |
| Age, mean (IQR) | 60.3 (54–68) | 62.0 (56–69) | 60.1 (54–67) | 0.19 | 61.7 | 61.7 | 0.00 |
| Male sex | 1992 (56.2) | 279 (57.4) | 1713 (56.0) | 0.03 | 56.5 | 56.5 | 0.00 |
| ADG score, mean (IQR) | 28.7 (22–36) | 28.9 (21–35) | 28.7 (22–36) | 0.02 | 28.9 | 28.9 | 0.00 |
| Material deprivation quintile | | | | | | | |
| 1 (least deprived) | 758 (21.4) | 121 (24.9) | 637 (20.8) | 0.10 | 23.4 | 23.4 | 0.00 |
| 2 | 708 (20.0) | 84 (17.3) | 624 (20.4) | 0.08 | 17.4 | 17.4 | 0.00 |
| 3 | 734 (20.7) | 100 (20.6) | 634 (20.7) | 0.00 | 20.5 | 20.5 | 0.00 |
| 4 | 714 (20.1) | 97 (20.0) | 617 (20.2) | 0.01 | 20.5 | 20.5 | 0.00 |
| 5 (most deprived) | 605 (17.1) | * 79–83 (16.3–17.1) | * 522–526 (17.1–17.2) | 0.01 | 17.5 | 17.5 | 0.00 |
| Missing | 27 (0.8) | * 1–5 (0.2–1.0) | * 22–26 (0.7–0.8) | 0.02 | 0.7 | 0.7 | 0.00 |
| Frail | 115 (3.2) | 22 (4.5) | 93 (3.0) | 0.08 | 4.4 | 4.4 | 0.00 |
| Rural residence | 477 (13.5) | 55 (11.3) | 422 (13.8) | 0.08 | 11.8 | 11.8 | 0.00 |
| High-risk (T4 or N2) | 1788 (50.4) | 123 (25.3) | 1665 (54.4) | 0.62 | 33.1 | 33.1 | 0.00 |
| Proximal tumor location (versus distal) | 1828 (51.6) | 238 (49.0) | 1590 (52.0) | 0.06 | 49.1 | 49.1 | 0.00 |
| Diagnosis year | | | | | | | |
| 2007–2011 | 1008 (28.4) | 52 (10.7) | 956 (31.2) | 0.52 | 14.4 | 14.4 | 0.00 |
| 2012–2015 | 1338 (37.7) | 123 (25.3) | 1215 (39.7) | 0.31 | 32.7 | 32.7 | 0.00 |
| 2016–2019 | 1200 (33.8) | 311 (64.0) | 889 (29.1) | 0.75 | 52.9 | 52.9 | 0.00 |
| Postoperative complication within 30 days of index operation | 1050 (29.6) | 148 (30.5) | 902 (29.5) | 0.02 | 29.3 | 29.3 | 0.00 |
| Diagnosis to surgery interval in days, mean (IQR) | 16.9 (0–29) | 18.6 (0–32) | 16.6 (0–29) | 0.09 | 18.5 | 18.5 | 0.00 |
| Surgery to adjuvant therapy interval in days, mean (IQR) | 50.8 (36–64) | 52.7 (36–67) | 50.5 (49.7–51.3) | 0.09 | 52.2 | 52.2 | 0.00 |
| FOLFOX (versus CAPOX) | 3186 (89.8) | 329 (67.7) | 2857 (93.4) | 0.69 | 80.2 | 80.2 | 0.00 |
| Dose reduction | 1236 (34.9) | 110 (22.6) | 1126 (36.8) | 0.31 | 26.9 | 26.9 | 0.00 |
| Chemotherapy complication | 1156 (32.6) | 153 (31.5) | 1003 (32.8) | 0.03 | 33.3 | 33.3 | 0.00 |

Abbreviation: ADG, Aggregated Diagnosis Groups. * Cells containing fewer than six individuals and adjacent cells are suppressed to mitigate re-identification risk.

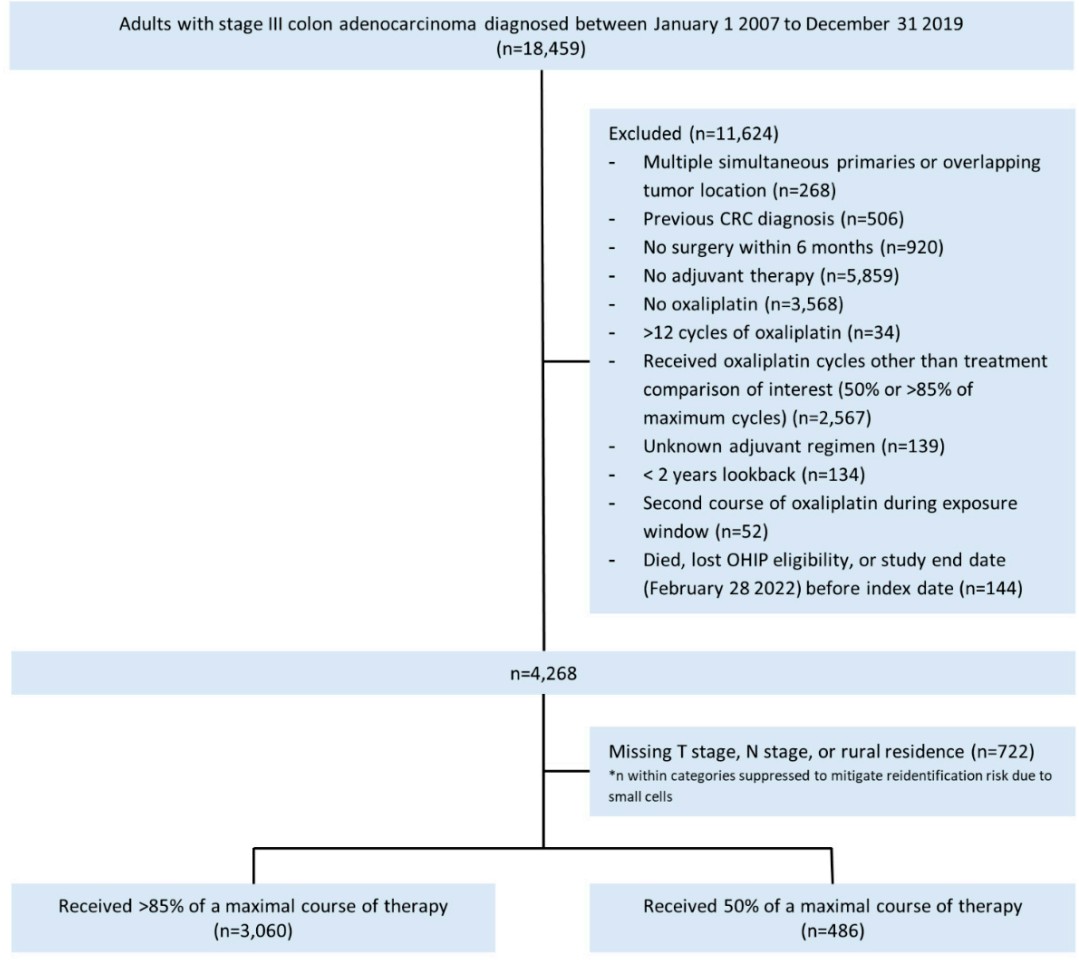

**Figure 1.** Study flow diagram for overall mortality in the primary comparison.

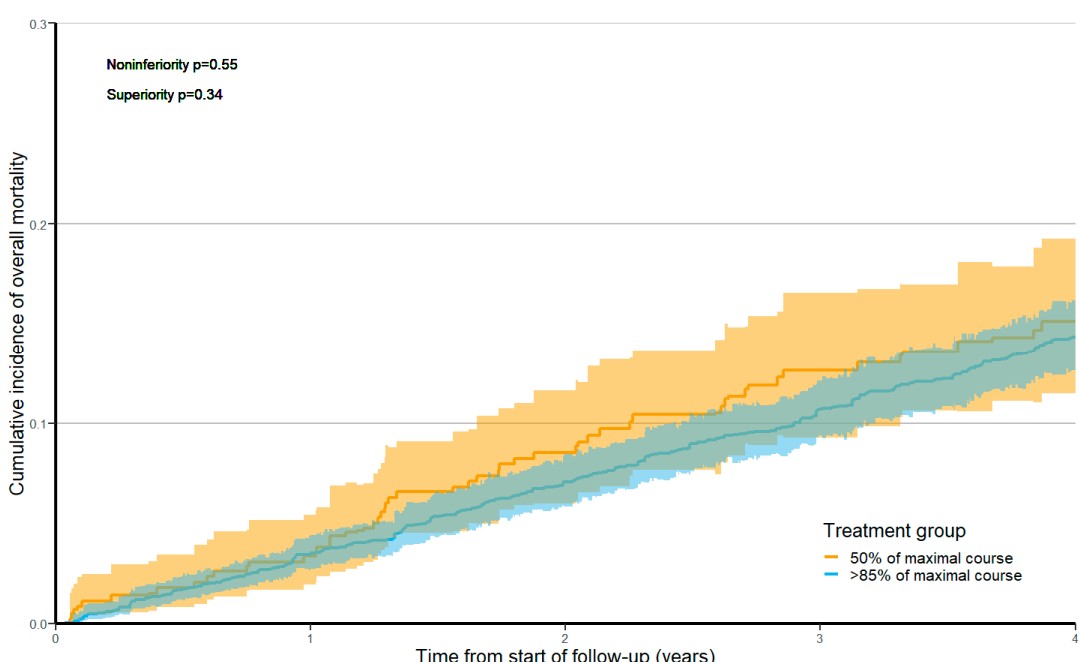

**Figure 2.** Cumulative incidence function for overall mortality stratified by treatment group after applying overlap weights. Shaded areas represent 95% confidence intervals. The time zero of follow-up begins 270 days after the first adjuvant treatment date.

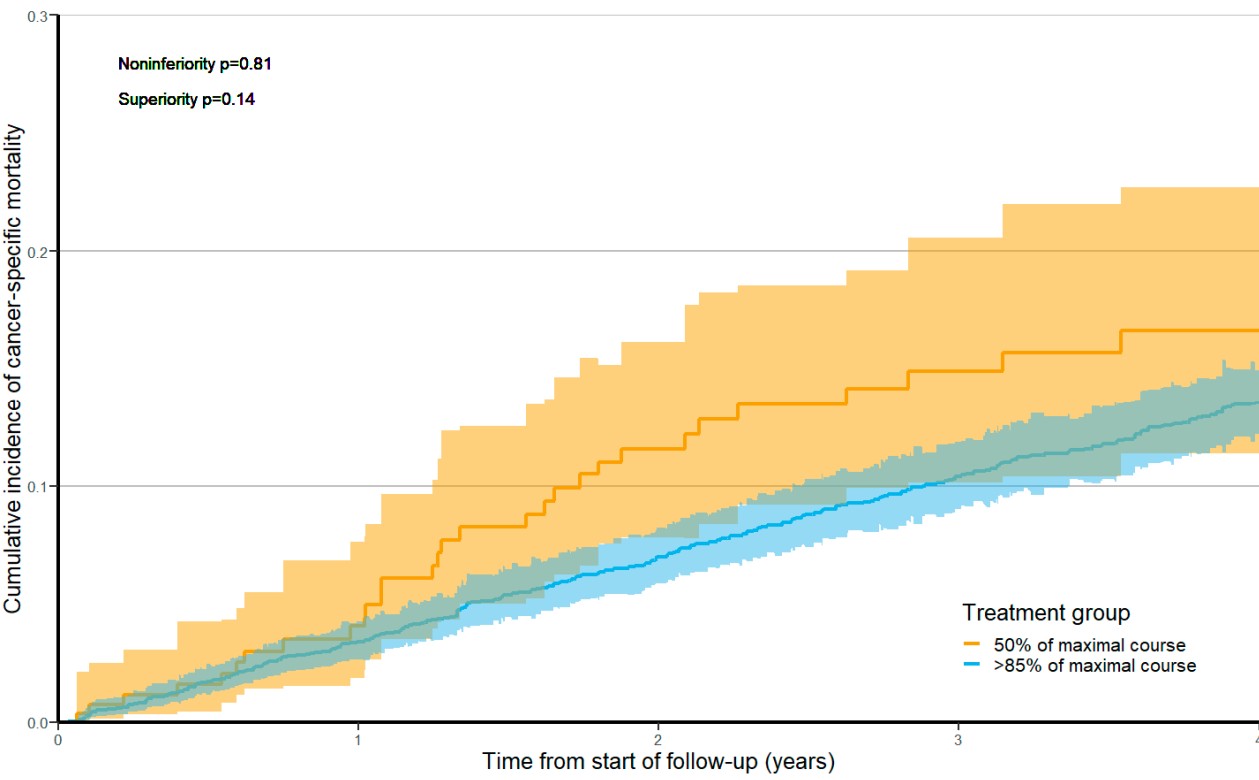

**Figure 3.** Cumulative incidence function for cancer-specific mortality stratified by treatment group, with non-cancer mortality as a competing risk and after applying overlap weights. Shaded areas represent 95% confidence intervals. Time zero of follow-up begins 270 days after the first adjuvant treatment date.

The cancer-specific mortality risk over 5 years was 17.6% (95% CI 12.5–23.4%) among patients treated with 50% of a maximal course of therapy versus 14.0% (95% CI 12.6–15.6%) among patients treated with >85% of a maximal course, where other-cause mortality was handled as a competing event. The sHR for the association of treatment with 50% versus >85% cycles of oxaliplatin-containing adjuvant chemotherapy and cancer-specific mortality was 1.31 (95% CI 0.91–1.87, non-inferiority $p = 0.81$, superiority $p = 0.14$).

Estimates of the association between treatment duration and mortality for the primary comparison within patient subgroups are presented in Figure 4. The HRs for overall mortality, comparing 50% against >85% of maximal therapy, were similar between high-risk (HR 1.11 [95% CI 0.81–1.54]) and low-risk cancers (HR 1.07 [95% CI 0.72–1.57]), as well as between CAPOX (HR 1.14 [95% CI 0.52–2.53]) and FOLFOX (HR 1.14 [95% CI 0.88 to 1.48]). The point estimates for the sHR for cancer-specific mortality between high-risk (sHR 1.30 [95% CI 0.87–1.94]) and low-risk cancers (0.79 [95% CI 0.36–1.76]) differed to a greater degree than the estimates for the HR for OS (Figure 5).

In the secondary comparison for the outcome of overall mortality, there may have been a trend toward increasing mortality with decreasing treatment duration—patients treated with seven cycles had a HR for overall mortality of 1.53 (95% CI 1.19 to 1.95) versus 1.07 (95% CI 0.87 to 1.32) for patients treated with ten cycles—but this was complicated by confidence intervals that overlapped across dose increments and a hazard ratio for six cycles that was below that of seven cycles (Table 1). For cancer-specific mortality, there was no clear trend across cycles.

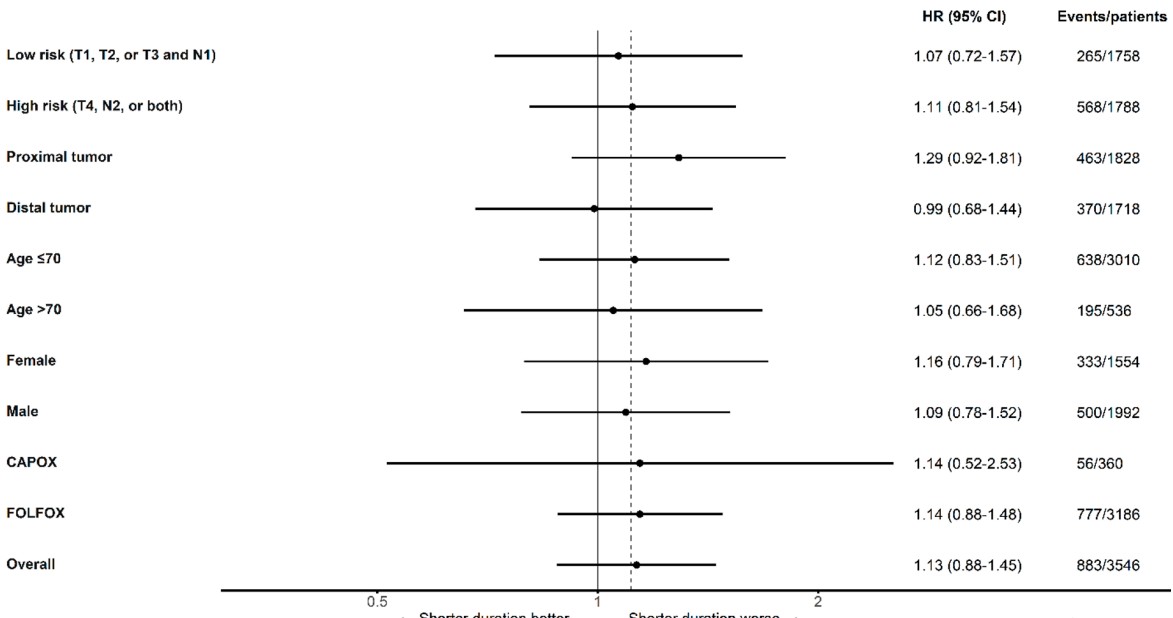

**Figure 4.** Overall mortality for 50% versus >85% of a maximum duration of oxaliplatin-containing adjuvant chemotherapy by subgroup after applying overlap weights. Dashed line denotes non-inferiority margin. CI, confidence interval; HR, hazard ratio. The finding that some subgroup estimates do not bound the overall estimate results from the non-collapsibility of hazard ratios obtained from our models. Because of this characteristic of hazard ratios, the overall estimate is not a weighted average of subgroup estimates, and both subgroup estimates may lie to either side of the overall estimate, even in the absence of confounding [35].

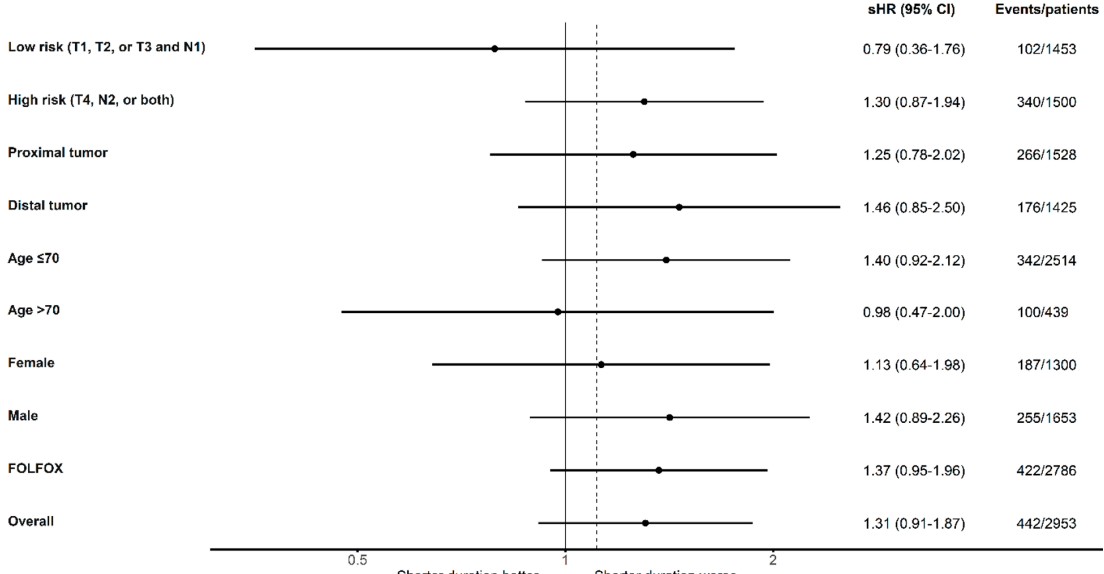

**Figure 5.** Cancer-specific mortality for 50% versus >85% of a maximum duration of oxaliplatin-containing adjuvant chemotherapy by subgroup after applying overlap weights. The dashed line denotes the non-inferiority margin. CI, confidence interval; sHR, subdistribution hazard ratio. The estimate for cancer-specific mortality within patients who received CAPOX is not shown because the propensity score was not estimable in this small subgroup. The finding that some subgroup estimates do not bound the overall estimate results from the non-collapsibility of hazard ratios obtained from our models. Because of this characteristic of hazard ratios, the overall estimate is not a weighted average of subgroup estimates, and both subgroup estimates may lie to either side of the overall estimate even in the absence of confounding factors [35].

*Sensitivity Analyses*

The results from multiple imputation, excluding patients with dose reductions and excluding diagnosis years after 2017, did not substantially differ from the results of the primary analysis (Table 4).

**Table 4.** Results from sensitivity analyses comparing 50% versus >85% of a maximal course of oxaliplatin-containing adjuvant therapy. Estimates are derived from overlap-weighted Cox and subdistribution hazard models.

| Sensitivity Analysis | HR (95% CI) for Overall Mortality | sHR (95% CI) for Cancer-Specific Mortality |
|---|---|---|
| Multiple imputation | 1.21 (0.98 to 1.50) | 1.41 (1.06 to 1.86) |
| Exclude patients with dose reductions | 1.19 (0.89 to 1.60) | 1.42 (0.94 to 2.15) |
| Exclude patients diagnosed after 2017 | 1.21 (0.93 to 1.57) | 1.18 (0.83 to 1.69) |

## 4. Discussion

In this population-based retrospective cohort study of patients with stage III colon cancer, we could not conclude that 50% is non-inferior to >85% of a maximal course of oxaliplatin-containing adjuvant therapy for overall or cancer-specific mortality. This finding was consistent across subgroups and several sensitivity analyses. For the main comparison between 50% and >85% of a maximal course of adjuvant therapy, our results for overall mortality were compatible with those from the IDEA collaboration, despite our cohort more often having high-risk tumors (50.4% versus 41.3%) and being treated with FOLFOX (89.8% versus 60.5%) [11]. Similar to the results of the IDEA study, our point estimate for the sHR for cancer-specific mortality showed a greater difference in mortality rates between treatment arms for patients with high-risk cancers than patients with low-risk cancers, though confidence intervals within these subgroups overlapped. In our secondary comparison to explore a dose effect, our estimates for both overall mortality and cancer-specific mortality had wide confidence intervals, but there was a trend toward increasing overall mortality rates relative to the longest-treated group as the duration of therapy decreased.

Our findings can be contrasted with previously published observational studies. A recent systematic review and superiority meta-analysis of 1258 patients did not find a statistically significant difference between treatment with 3 versus 6 months of doublet adjuvant therapy, but several of the included observational studies may have suffered from immortal time bias [36].

After the publication of this review, two population-based retrospective cohort studies sought to better define the association of shorter adjuvant therapy with survival, while accounting for immortal time bias. The first used a superiority approach to examine the association between treatment with 3–5 months versus 6 months of oxaliplatin-containing adjuvant chemotherapy and all-cause mortality [37]. Mortality rates were similar between groups. However, the short-duration therapy in the study significantly exceeded the duration of treatment received by the short-duration group in IDEA, who received only 3 months. Conversely, our comparator groups were chosen to reflect the treatment decision clinicians face between terminating treatment at 3 months and completing the full course.

The second examined the association between treatment with varying durations of oxaliplatin-containing adjuvant chemotherapy and cancer-specific mortality within the strata of a regimen [38]. In this study, FOLFOX for twelve cycles and CAPOX for eight cycles were superior to incomplete courses of either therapy. Notably, they included patients who completed as few as one cycle of adjuvant therapy in their shortest treatment comparison group. Patients who only tolerate one cycle of adjuvant therapy are likely significantly more unwell at baseline than patients who can tolerate even 50% of a maximal course, increasing the risk of confounding by indication in this population, and potentially explaining our differing findings.

Like these two recently published observational studies, our study leveraged high-quality linked databases to access a large, population-based sample, allowing for numerous clinically important subgroup analyses and complete follow-up. Our study had several additional strengths in its design, analysis, and interpretation. We accounted for baseline between-group differences using overlap weights to obtain an exact balance of measured covariates, allowing us to target a clinically relevant estimate among patients most similar to one another. We formally tested for non-inferiority and framed our findings using non-inferiority language to explicitly account for the crucial non-efficacy benefits of shortened therapy. We identified treatment and comparator groups having received similar proportions of a maximal course of adjuvant therapy as those in the IDEA collaboration, and examined for a dose effect. Our analysis of a dose effect, while far from conclusive, should encourage further head-to-head comparisons of treatment durations between and beyond those examined in IDEA.

Our study has several limitations. First, our sHR estimates for cancer-specific mortality lacked precision because cause of death data were available only to the end of 2017, limiting our follow-up and event counts for this outcome. Second, while available fluoropyrimidine data allowed us to identify patients receiving 5-FU versus capecitabine, chemotherapy administration dates and doses were reliably captured only for oxaliplatin. Our exposure thus corresponds most closely to oxaliplatin. Third, despite our large sample size, most patients received FOLFOX in our cohort, resulting in imprecise estimates for CAPOX that are difficult to compare with those from IDEA. Lastly, our routinely collected data did not allow us to ascertain some variables related to both exposure and outcome (e.g., neuropathy, dose intensity, smoking status, and cancer recurrence during the treatment window) [39,40]. Patients who stopped treatment early in the pre-IDEA period may have done so for reasons related to these variables, raising the concern that our results are biased in favor of long-duration therapy in the absence of control for these variables. Given the limitations of our study design, residual confounding cannot be ruled out as an explanation for our findings. However, our analysis captured all important confounders and most potential confounders identified by a recent systematic review of adjuvant duration for stage III colon cancer [36]. Moreover, the results of our sensitivity analysis carried out in the subset of patients who did not experience a dose reduction and were likely more similar to one another did not show a meaningful difference from our main results.

## 5. Conclusions

In this population-based retrospective cohort study of patients treated with oxaliplatin-containing adjuvant chemotherapy for stage III colon cancer, we could not conclude that treatment with 50% of a maximal course of adjuvant therapy is associated with non-inferior mortality compared to treatment with >85% of a maximal course in routine practice. The tradeoff between mortality and treatment toxicity remains important to consider carefully in shared decision-making around adjuvant duration, regardless of risk subgroup.

**Author Contributions:** Conceptualization: C.S.-C.-L. and N.N.B.; Data curation: C.S.-C.-L.; Formal analysis: C.S.-C.-L.; Funding acquisition: C.S.-C.-L. and N.N.B.; Methodology: C.S.-C.-L., C.B.-M., R.S., A.Y.X.Y. and N.N.B.; Supervision: N.N.B.; Writing—original draft: C.S.-C.-L.; Writing—review and editing: C.B.-M., R.S., A.Y.X.Y. and N.N.B. All authors have read and agreed to the published version of the manuscript.

**Funding:** This study was supported by ICES, which is funded by an annual grant from the Ontario Ministry of Health (MOH) and the Ministry of Long-Term Care (MLTC). Funding for this study was also provided by the PSI Foundation and a CIHR Foundation Grant (#148470). Parts of this material are based on data and information compiled and provided by the Statistics Canada Postal CodeOM Conversion File, which is based on data licensed from Canada Post Corporation, and/or data adapted from the Ontario Ministry of Health Postal Code Conversion File, which contains data copied under license from Canada Post Corporation and Statistics Canada; Ontario Health; the Ontario Ministry of Health; the Canadian Institute for Health Information; the Ontario Community Health Profiles Partnership; and the Ontario Registrar General (the original source of which is ServiceOntario). The

opinions, results, view, and conclusions expressed herein are solely those of the authors and do not reflect those of the funding or data sources; no endorsement is intended or should be inferred.

**Institutional Review Board Statement:** The Research Ethics Board at St. Michael's Hospital (#20-256) and the Office of Research Ethics at the University of Toronto (#40355) provided approval for the present study.

**Informed Consent Statement:** The Research Ethics Board at St. Michael's Hospital (#20-256) and the Office of Research Ethics at the University of Toronto (#40355) waived the requirement for informed consent for this work. ICES is a prescribed entity under Ontario's Personal Health Information Protection Act (PHIPA). Section 45 of PHIPA authorizes ICES to collect personal health information, without consent, for the purpose of analysis or compiling statistical information with respect to the management of, evaluation or monitoring of, the allocation of resources to or planning for all or part of the health system.

**Data Availability Statement:** The dataset from this study is held securely in coded form at ICES. While legal data sharing agreements between ICES and data providers (e.g., healthcare organizations and government) prohibit ICES from making the dataset publicly available, access may be granted to those who meet pre-specified criteria for confidential access, available at www.ices.on.ca/DAS (accessed on 3 July 2023) (email: das@ices.on.ca). The full dataset creation plan and underlying analytic code are available from the authors upon request, understanding that the computer programs may rely upon coding templates or macros that are unique to ICES and are therefore either inaccessible or may require modification.

**Acknowledgments:** We thank Sho Podolsky and Ning Liu for their assistance in obtaining and preparing the data. We thank IQVIA Solutions Canada Inc. for use of their Drug Information File.

**Conflicts of Interest:** Brezden has received grants from Hoffman-La Roche, Novartis, Eli Lilly, Amgen, Pfizer, Taiho, Astra Zeneca, and Astellas; honoraria from Hoffman-La Roche, Novartis, Eli Lilly, Amgen, Pfizer, Taiho, Gilead Sciences, Agendia, and Seagen; and accommodations from Hoffman-La Roche, Novartis, Eli Lilly, Amgen, Pfizer, Taiho, Gilead Sciences, Agendia, and Seagen outside the submitted work. All other authors have no competing interest.

## Appendix A

**Table A1.** The RECORD statement, a checklist of items, extended from the STROBE statement, should be reported in observational studies using routinely collected health data [10].

| | Item No. | STROBE Items | Location in Manuscript Where Items Are Reported | RECORD Items | Location in Manuscript Where Items Are Reported |
|---|---|---|---|---|---|
| **Title and abstract** | | | | | |
| | 1 | (a) Indicate the study's design with a commonly used term in the title or the abstract (b) Provide in the abstract an informative and balanced summary of what was done and what was found | | RECORD 1.1: The type of data used should be specified in the title or abstract. When possible, the name of the databases used should be included.<br><br>RECORD 1.2: If applicable, the geographic region and timeframe within which the study took place should be reported in the title or abstract.<br><br>RECORD 1.3: If linkage between databases was conducted for the study, this should be clearly stated in the title or abstract. | Page 1–2 |
| **Introduction** | | | | | |
| Background rationale | 2 | Explain the scientific background and rationale for the investigation being reported | | | Page 2 |
| Objectives | 3 | State specific objectives, including any prespecified hypotheses | | | Page 2 |
| **Methods** | | | | | |
| Study Design | 4 | Present key elements of the study design early in the paper | | | Page 2–5 |

**Table A1.** *Cont.*

| | Item No. | STROBE Items | Location in Manuscript Where Items Are Reported | RECORD Items | Location in Manuscript Where Items Are Reported |
|---|---|---|---|---|---|
| Setting | 5 | Describe the setting, locations, and relevant dates, including periods of recruitment, exposure, follow-up, and data collection | | | Page 2–5 |
| Participants | 6 | (a) *Cohort study*—Give the eligibility criteria, and the sources and methods of selection of participants. Describe methods of follow-up *Case-control study*—Give the eligibility criteria, and the sources and methods of case ascertainment and control selection. Give the rationale for the choice of cases and controls *Cross-sectional study*—Give the eligibility criteria, and the sources and methods of selection of participants<br><br>(b) *Cohort study*—For matched studies, give matching criteria and number of exposed and unexposed *Case-control study*—For matched studies, give matching criteria and the number of controls per case | | RECORD 6.1: The methods of study population selection (such as codes or algorithms used to identify subjects) should be listed in detail. If this is not possible, an explanation should be provided.<br><br>RECORD 6.2: Any validation studies of the codes or algorithms used to select the population should be referenced. If validation was conducted for this study and not published elsewhere, detailed methods and results should be provided.<br><br>RECORD 6.3: If the study involved linkage of databases, consider use of a flow diagram or other graphical display to demonstrate the data linkage process, including the number of individuals with linked data at each stage. | Page 2–5, Appendix A |
| Variables | 7 | Clearly define all outcomes, exposures, predictors, potential confounders, and effect modifiers. Give diagnostic criteria, if applicable. | | RECORD 7.1: A complete list of codes and algorithms used to classify exposures, outcomes, confounders, and effect modifiers should be provided. If these cannot be reported, an explanation should be provided. | Page 2–5, Appendix A |

| | Item No. | STROBE Items | Location in Manuscript Where Items Are Reported | RECORD Items | Location in Manuscript Where Items Are Reported |
|---|---|---|---|---|---|
| Data sources/measurement | 8 | For each variable of interest, give sources of data and details of methods of assessment (measurement). Describe comparability of assessment methods if there is more than one group | | | Page 2–5, Appendix A |
| Bias | 9 | Describe any efforts to address potential sources of bias | | | Page 2–5 |
| Study size | 10 | Explain how the study size was arrived at | | | Figure 1 |
| Quantitative variables | 11 | Explain how quantitative variables were handled in the analyses. If applicable, describe which groupings were chosen, and why | | | Page 2–5 |
| Statistical methods | 12 | (a) Describe all statistical methods, including those used to control for confounding (b) Describe any methods used to examine subgroups and interactions (c) Explain how missing data were addressed (d) *Cohort study*—If applicable, explain how loss to follow-up was addressed *Case-control study*—If applicable, explain how matching of cases and controls was addressed *Cross-sectional study*—If applicable, describe analytical methods taking account of sampling strategy (e) Describe any sensitivity analyses | | | Page 2–5 |

**Table A1.** *Cont.*

| | Item No. | STROBE Items | Location in Manuscript Where Items Are Reported | RECORD Items | Location in Manuscript Where Items Are Reported |
|---|---|---|---|---|---|
| Data access and cleaning methods | | .. | | RECORD 12.1: Authors should describe the extent to which the investigators had access to the database population used to create the study population.<br><br>RECORD 12.2: Authors should provide information on the data cleaning methods used in the study. | Page 2–5 |
| Linkage | | .. | | RECORD 12.3: State whether the study included person-level, institutional-level, or other data linkage across two or more databases. The methods of linkage and methods of linkage quality evaluation should be provided. | Page 2–5 |
| **Results** | | | | | |
| Participants | 13 | (a) Report the numbers of individuals at each stage of the study (e.g., numbers potentially eligible, examined for eligibility, confirmed eligible, included in the study, completing follow-up, and analyzed)<br>(b) Give reasons for non-participation at each stage<br>(c) Consider use of a flow diagram | | RECORD 13.1: Describe in detail the selection of the persons included in the study (i.e., study population selection) including filtering based on data quality, data availability and linkage. The selection of included persons can be described in the text and/or by means of the study flow diagram. | Page 5, Figure 1 |

**Table A1.** *Cont.*

| | Item No. | STROBE Items | Location in Manuscript Where Items Are Reported | RECORD Items | Location in Manuscript Where Items Are Reported |
|---|---|---|---|---|---|
| Descriptive data | 14 | (a) Give characteristics of study participants (e.g., demographic, clinical, social) and information on exposures and potential confounders (b) Indicate the number of participants with missing data for each variable of interest (c) *Cohort study*—summarize follow-up time (e.g., average and total amount) | | | Page 5, Table 1 |
| Outcome data | 15 | *Cohort study*—Report numbers of outcome events or summary measures over time *Case-control study*—Report numbers in each exposure category, or summary measures of exposure *Cross-sectional study*—Report numbers of outcome events or summary measures | | | Page 7–8 |
| Main results | 16 | (a) Give unadjusted estimates and, if applicable, confounder-adjusted estimates and their precision (e.g., 95% confidence interval). Make clear which confounders were adjusted for and why they were included (b) Report category boundaries when continuous variables were categorized (c) If relevant, consider translating estimates of relative risk into absolute risk for a meaningful time period | | | Page 8–9 |

**Table A1.** *Cont.*

| | Item No. | STROBE Items | Location in Manuscript Where Items Are Reported | RECORD Items | Location in Manuscript Where Items Are Reported |
|---|---|---|---|---|---|
| Other analyses | 17 | Report other analyses performed—e.g., analyses of subgroups and interactions, and sensitivity analyses | | | Page 9–11 |
| **Discussion** | | | | | |
| Key results | 18 | Summarize key results with reference to study objectives | | | Page 11–13 |
| Limitations | 19 | Discuss limitations of the study, taking into account sources of potential bias or imprecision. Discuss both direction and magnitude of any potential bias | | RECORD 19.1: Discuss the implications of using data that were not created or collected to answer the specific research question(s). Include discussion of misclassification bias, unmeasured confounding, missing data, and changing eligibility over time, as they pertain to the study being reported. | Page 12–13 |
| Interpretation | 20 | Give a cautious overall interpretation of results considering objectives, limitations, multiplicity of analyses, results from similar studies, and other relevant evidence | | | Page 11–13 |
| Generalizability | 21 | Discuss the generalizability (external validity) of the study results | | | Page 11–13 |
| **Other Information** | | | | | |
| Funding | 22 | Give the source of funding and the role of the funders for the present study and, if applicable, for the original study on which the present article is based | | | Page 14 |

**Table A1.** *Cont.*

| | Item No. | STROBE Items | Location in Manuscript Where Items Are Reported | RECORD Items | Location in Manuscript Where Items Are Reported |
|---|---|---|---|---|---|
| Accessibility of protocol, raw data, and programming code | | .. | | RECORD 22.1: Authors should provide information on how to access any supplemental information such as the study protocol, raw data, or programming code. | Page 14 |

**Table A2.** Coding algorithms for cohort definition, exposure, outcome, and patient characteristics.

| Characteristic | Data Source | Codes |
| --- | --- | --- |
| Sex | RPDB | RPDB SEX = M or F |
| ADG Score [12] | NACRS, DAD, OHIP | ADG comorbidity score derived from weighted ADG categories present during 2-year lookback at outpatient and inpatient records |
| Frailty | NACRS, DAD, OHIP | ACG Flag FRAILTY = yes |
| Adjuvant regimen | NDFP, ALR, OHIP, ODB | Hierarchical algorithm criteria:<br>1. Modal cycle interval by NDFP: if 18–24 days between cycles, then CAPOX; if 11–17 days between cycles then FOLFOX<br>2. First cycle interval by NDFP: if 18–24 days between cycles, then CAPOX; if 11–17 days between cycles then FOLFOX<br>3. Modal oxaliplatin-containing regimen by ALR<br>4. First oxaliplatin-containing regimen by ALR<br>5. If received oxaliplatin and OHIP billing code G388 for oral chemotherapy during exposure window, then CAPOX<br>6. If received oxaliplatin and ODB claim for oral chemotherapy (DIN 02426765, 02457504, 02421917, 02457490, 02426757, 02400022, 02238453, 02421925, 02400030, 02238454) during exposure window, then CAPOX |
| Postoperative complication within 30 days of index operation [19] | DAD, NACRS, OHIP | Reoperation for intra-abdominal complication<br>CCI<br>1.NK.80.^^,1.NM.52.^^,1.NM.80.^^,1.NP.86.^^,1.OT.13.^,1.OT.52.^^,1.OT.70.LA,1.NK.76.^^,1.NK.77. ^^,1.NK.87.^^,1.NM.76.^^,1.NM.77.^^,1.NM.87.^^,1.NM.89.^^,1.NM.91.^^<br><br>Venous thromboembolism or pulmonary embolism<br>ICD10<br>I.26.^^,I.80.1-I.80.3<br><br>Sepsis<br>ICD10<br>A.41.^^,A.41.1.,A.41.2.,A.41.3.,A.41.4.,A.41.5.^^,A.41.8.^^,A.41.9.<br><br>Hemorrhage<br>CCI<br>1.LZ.19^^<br>ICD10<br>T.81.0,T81.1,R.58.<br><br>Percutaneous drainage of abdominal abscess<br>OHIP<br>S313,S314,Z569,Z594<br><br>Major wound disruption<br>CCI<br>1.SY.80^^<br>ICD10<br>T.81.3.<br>OHIP<br>S343<br><br>Fistula formation<br>CCI<br>1.NP.86.^^<br>ICD10<br>K.63.2.,K.31.6.,N.32.1.<br>OHIP<br>E714<br><br>Wound infection<br>ICD10<br>T.81.4.<br><br>Stroke or transient ischemic attack<br>ICD10 G.45.^^,I.60.^^,I.61.^^,I.63.^^,I.64.,H.34.1<br><br>Myocardial infarction<br>ICD10 I.21.^^,I.22.^^,I.23.^^<br><br>Congestive Heart Failure<br>ICD10 I.50.^^ |

**Table A2.** *Cont.*

| Characteristic | Data Source | Codes |
|---|---|---|
| Dose reduction | NDFP | Any oxaliplatin dose <80% of the first dose [37] |
| Chemotherapy complication requiring ED visit or hospital admission [41–44] | DAD, NACRS | ICD10 in any diagnostic space<br>Neutropenia<br>Agranulocytosis (D70.*)<br>Fever<br>Fever of unknown origin (R50.*)<br>Infection<br>Infectious and parasitic disease, including sepsis (A00.*–B99.*)<br>Infection and inflammatory reaction due to other cardiac and vascular devices, implants and grafts (T82.7)<br>Bronchitis (J20.*–J22.*)<br>Pneumonia (J09.*–J11.*)<br>Kidney infection (N10, N39.0)<br>Acute cystitis (N30.0)<br>Cellulitis (L00.*–L08.*)<br>Empyema (J86.*)<br>Abscess lung/mediastinum (J85.*)<br>GI Toxicity<br>Diarrhea, colitis (K52.*)<br>Functional diarrhea (K59.1)<br>Nausea and vomiting (R11.*)<br>Heartburn (R12.*)<br>Constipation (K59.0)<br>Obstruction (includes ileus) (K56.*)<br>Stomatitis (K12.*)<br>Cachexia (R64.*)<br>Anorexia (R63.0)<br>Other systemic treatment-related<br>Hyponatremia (E87.1)<br>Hypokalemia (E87.6)<br>Other electrolyte/fluid abnormality (E87.*)<br>Magnesium disorder (E83.4)<br>Dehydration/hypovolemia (E86.*)<br>Malaise/fatigue (R53.*)<br>Syncope (R55.*)<br>Dizziness (R42.*)<br>Hypotension (I95.9)<br>Fe deficiency anemia (D50.*)<br>Other deficiency anemia (D51.*–D53.*)<br>Aplastic anemia (D60.*–D61.*)<br>Other and unspecified anemia (D62.*–D64.*)<br>Thrombocytopenia (D69.5, D69.6)<br>Other venous embolism and thrombosis (I82.*)<br>Rash and non-specific skin eruptions (R21.*)<br>Hyperglycemia (R73.*)<br>Phlebitis and thrombophlebitis (I80.*)<br>Pulmonary embolism (I26.*)<br>Disorders of calcium metabolism (E83.5)<br>Disorders of phosphorus metabolism and phosphatases (E83.3) |
| Deprivation quintile | CENSUS | %GETONMARG macro variable deprivation_q_da, based on most recent dissemination area prior to first adjuvant treatment date |
| Rurality | CENSUS | binary variable rural = 1 if PCCF rural flag = Y |
| AJCC Stage | OCR | BEST_STAGE_GRP |
| Cause of death | ORGD | Cancer-specific mortality: ICD9 14–23 |
| Oxaliplatin | NDFP | DRUG_NAME = 'Oxaliplatin' |
| Colon cancer diagnosis | OCR | Proximal colon: ICD-O-3 Topography code C180, C182-C184<br>Distal colon: ICD-O-3 Topography code C185-C187, C199 [18] |
| Colon resection | DAD | CCI 1NM76, 1NM77, 1NM87, 1NM89, 1NM91, 1NQ87, 1NQ89 [45–47] |

Abbreviations: ADG, Aggregated Diagnosis Group; AJCC, American Joint Committee on Cancer; ALR, Activity Level Registry; CCI, Canadian Classification of Interventions; DAD, Discharge Abstract Database; NACRS, National Ambulatory Care Reporting System; NDFP, New Drug Funding Program; OCR, Ontario Cancer Registry; OHIP, Ontario Health Insurance Plan; RPDB, Registered Persons' Database. * indicates a wildcard/any value.

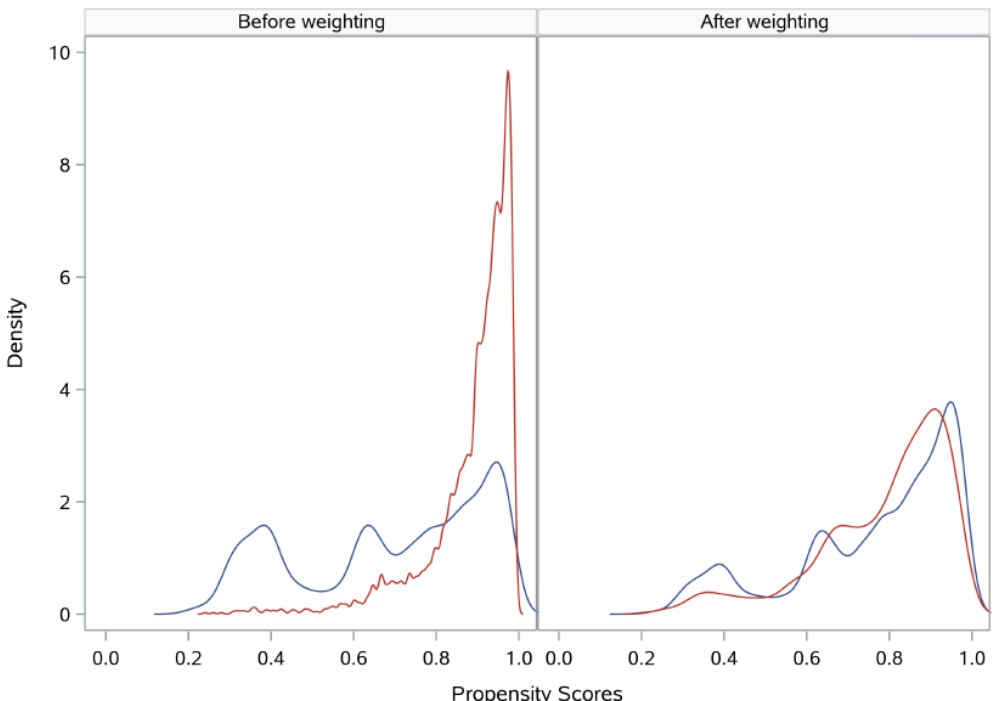

**Figure A1.** Propensity score distribution before and after weighting for patients analyzed for overall mortality. Blue indicates treatment with 50% and red indicates treatment with >85% of a maximal course of therapy.

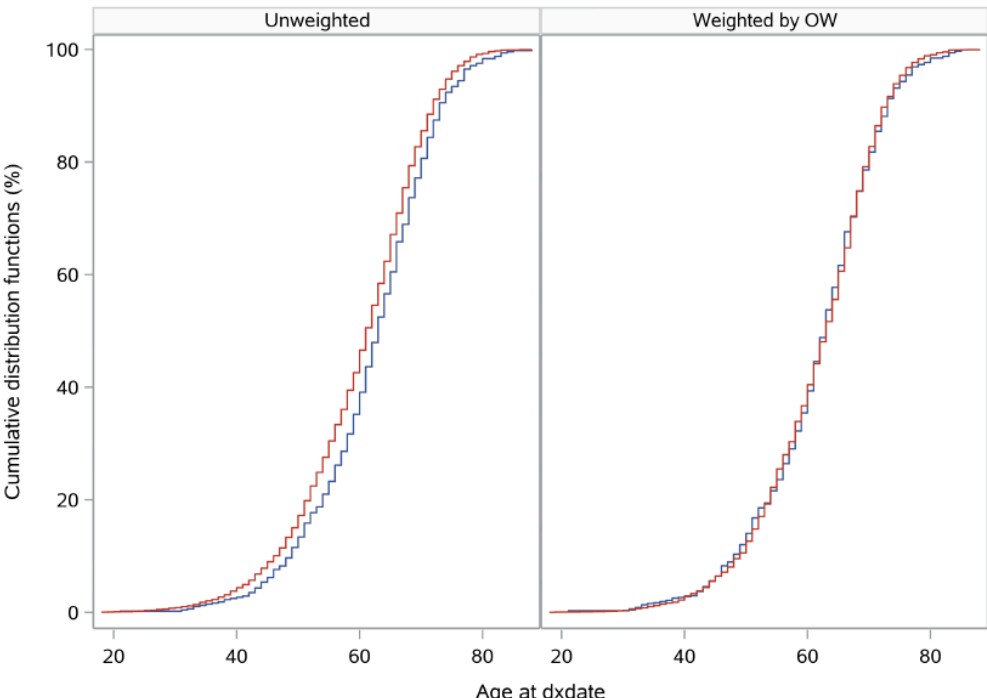

**Figure A2.** Cumulative distribution function for age before and after weighting for patients analyzed for overall mortality. Blue indicates treatment with 50% and red indicates treatment with >85% of a maximal course of therapy.

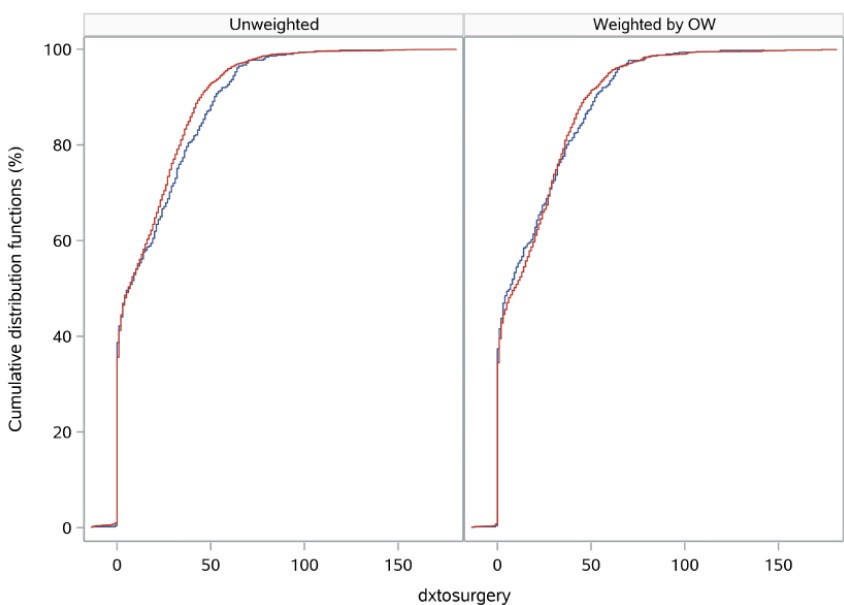

**Figure A3.** Cumulative distribution function for the interval in days between diagnosis and surgery before and after weighting for patients analyzed for overall mortality. Blue indicates treatment with 50% and red indicates treatment with >85% of a maximal course of therapy.

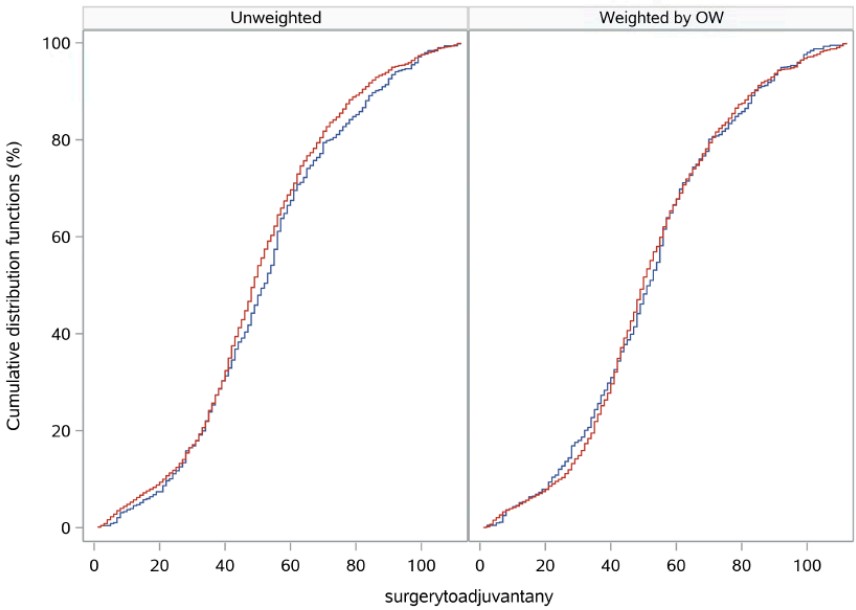

**Figure A4.** Cumulative distribution function for the interval in days between surgery and initiation of adjuvant therapy before and after weighting for patients analyzed for overall mortality. Blue indicates treatment with 50% and red indicates treatment with >85% of a maximal course of therapy.

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
