# Peer review of "The Association of Oxaliplatin-Containing Adjuvant Chemotherapy Duration with Overall and Cancer-Specific Mortality in Individuals with Stage III Colon Cancer: A Population-Based Retrospective Cohort Study"

_curroncol, doi:10.3390/curroncol30070478_

Round 1
Reviewer 1 Report
The authors present a well written manuscript on population level data for patients receiving adjuvant doublet chemotherapy for resected stage III colon cancer.
Would a patient with an early metastatic recurrence, after surgery, receiving 6 months of FOLFOX be flagged as having advanced disease? If they did not receive bevacizumab, EGFR mAb, or other therapies within the 270 day window? This needs to be noted in limitations if it is true.
The consort diagram is missing the 7,8,9, 10 cycle patients. This needs to be corrected. This group neither fits into the >85% or 50% group.
Despite the limitations of this data being population level, and prone to the biases that healthier patients would get more chemotherapy there are so notable take-aways. Lower risk patients treated with CAPOX do not appear to derive benefit from a longer duration, in keeping with the IDEA results.
Younger and higher risk patients appear to derive more benefit from a longer duration of therapy.
Median follow-up for 50% vs 85% groups should be given (required). 50% dosing is more common 2016-2019, so this median follow-up and cancer specific follow-up needs to be explicitly provided, as it may represent a significant limitation to this work. Given your limitation on causes of death to 2017 - this really needs to be provided.
Questions/limitations: why was diabetes not examined? Would ICES data not have indicated patients on diabetic medications, etc. Neuropathy with oxaliplatin is a major complication and a limiting factor in duration and dose intensity.
Line 304 to 307: The outcomes for patients getting 7 cycles appeared inferior, but 50% compared to 85% did not look as compromised. Do you suspect the reason patients stopped after 50% was different than those going for 7 cycles of FOLFOX (recurrence after a 3 month CT, performance status, smaller numbers, etc.)
Line 312 - 315: Can you make that conclusion when this did not reach significance? Wording should reflect the statistics.
Limitations: Dose intensity and time from surgery to initiation of adjuvant chemotherapy are established factors in recurrence. These were not part of the analysis and should be noted as a limitation - with citations added.
The conclusion is better wording than line 312 - 315, I think the conclusion is strong and an excellent summary.
Reviewer 2 Report
I suggest to include data about pack-years and smoking exposure
please include a section about inclusion and exclusion criteria
The equipe and technique used to stage colon cancer sholud be specified
FOLFOX and CAPOX schedules should be specified
